# Factors Associated with the Work Engagement of Employees Working from Home during the COVID-19 Pandemic in Japan

**DOI:** 10.3390/ijerph181910495

**Published:** 2021-10-06

**Authors:** Hoichi Amano, Yoshiharu Fukuda, Katsuhiko Shibuya, Akihiko Ozaki, Takahiro Tabuchi

**Affiliations:** 1Graduate School of Public Health, Teikyo University, Tokyo 173-8605, Japan; fukuday@med.teikyo-u.ac.jp (Y.F.); k-shibuya@med.teikyo-u.ac.jp (K.S.); 2Department of Breast Surgery, Jyoban Hospital of Tokiwa Foundation, Iwaki 972-8322, Japan; ozakiakihiko@gmail.com; 3Cancer Control Center, Osaka International Cancer Institute, Osaka 541-8567, Japan; tabuti-ta@mc.pref.osaka.jp

**Keywords:** COVID-19, work engagement, work from home

## Abstract

This study aimed to identify factors influencing the work engagement of employees working from home during the COVID-19 pandemic in Japan. Employees’ work engagement was examined using the following survey questions: “Do you feel energized when you are at work? (yes or no)” and “Do you take pride in your work? (yes or no)” After adjusting for potential confounders, Poisson regression was used to examine prevalence ratio and 95% confidence intervals for employees’ work engagement. We analyzed 15,670 individuals (11,894 of whom did not work from home and 3776 of whom worked from home). Their mean age was 45.6 ± 13.8 years, and 58.3% were men. Those who worked from home were younger than those who did not (43.9 ± 13.1 vs. 46.1 ± 13.9, *p* < 0.001). About 44% of all employees reported high work engagement. Among the employees who worked from home, an increase in sleep hours, effective interactions with supervisors, and working hours of ≤40 h/week were associated with engagement. Sensitivity analysis showed similar results. Close communication with superiors, refraining from working long hours, and obtaining adequate sleep may boost the work engagement of employees working from home.

## 1. Introduction

In 2020, the coronavirus disease 2019 (COVID-19) pandemic changed the way we live and work. As a result, today, there is a growing global trend to encourage more people to work from their homes [1,2]. Employees working from home face some of the biggest challenges, such as difficulties in achieving work/life balance, maintaining workspaces, collaborating and/or communicating with supervisors/colleagues, and adhering to healthy behaviors [3,4,5,6].

In recent years, occupational health researchers and health providers have increasingly focused on work engagement. Work engagement is defined as “a positive, fulfilling, work-related state of mind characterized by vigor, dedication, and absorption” [7]. In this definition, “vigor” is characterized by high levels of energy, the willingness to invest effort in one`s work, and persistence in working even in the face of difficulties. “Dedication” refers to being strongly involved in one`s work and experiencing a sense of significance, enthusiasm, inspiration, pride, and challenge. Finally, “absorption” is characterized by completely concentrating and being engrossed in one’s work, whereby time passes quickly. According to the Job Demands-Resource (JD-R) model, work engagement is the product of the balance between job demands (i.e., workload and time pressure) and available resources (i.e., decision latitude, supervisor support, co-worker support, and extrinsic reward) [8]. Furthermore, because many studies have examined work engagement as a predictor of job performance as well as employee health condition [8,9,10,11], work engagement improvement is considered one of the targets of workplace health promotion [12].

Accumulating evidence indicates that heavy workload, time pressure, and insufficient support from supervisors and coworkers are threats to work engagement [8]. In the wake of the COVID-19 pandemic, the number of employees working from home is increasing, and the factors affecting their work engagement are unknown. The extended working hours and difficulties in obtaining supervisory support, both of which are caused by the change in work styles following the pandemic, may negatively affect work engagement and lead to a decrease in productivity and an increase in health-related issues. Although work engagement improvement is considered one of the targets of workplace health promotion [12], there is insufficient understanding of the impact of the spread of COVID-19 on employees’ engagement in accordance with the new work style, i.e., working from home. Using the data from a large Internet survey conducted between August 25 and September 30, 2020 in Japan, we identified risk factors influencing employees’ work engagement among those working from home during the COVID-19 pandemic in Japan. Based on the JR model, we hypothesized that among employees working from home, those who did not have good communication with their supervisor or worked longer hours may have lower work engagement.

## 2. Materials and Methods

### 2.1. Data Collection

We analyzed data from the Japan “COVID-19 and Society” Internet Survey (JACSIS). The JACSIS is a large-scale, internet-based, self-reported questionnaire survey via a survey panel provided by a major internet survey agency in Japan (Rakuten Insight, Inc., Tokyo, Japan). The total number of individuals included in the survey panel was approximately 2.2 million and comprises individuals from diverse socioeconomic backgrounds—such as educational level, household income and number of household members, and marital status—to be nationally representative [13]. This study reached 224,389 participants using a stratified sampling approach by gender, age, and each prefecture from the panel. The study enrollment continued until it achieved the target numbers of respondents whose age, gender, and prefectures had been a priori set (based on the distribution of the general Japanese population in 2019 and 28,000 respondents). The survey was conducted between 25 August and 30 September 2020. The overall response rate was 12.5% (28,000/224,389). This study excluded respondents whose answers were inconsistent (specific items were included to identify inconsistent responses in the survey).

### 2.2. Assessment of Employees’ Work Engagement

To clarify employees’ work engagement, the survey included the following two questions: (1) “Do you feel energized when you are at work?” and (2) “Do you take pride in your work?” These questions are part of the New Brief Job Stress Questionnaire (New BJSQ) and are generally used to perform a “stress check” for the primary prevention of mental health issues [14,15]. In Japan, based on the Industrial Safety and Health Act, a “stress check” should be performed. In the New BJSQ, these two questions are used to evaluate employees’ work engagement because the two items measure different dimensions of this concept (i.e., vigor and dedication). Previous studies have shown that the New BJSQ scales are reliable and valid and that the New BJSQ is a useful instrument to evaluate the psychosocial work environment and positive mental health outcomes in the workplace [14,15]. Whereas the first question examined the main outcome, the second was used for sensitivity analysis. The permitted options for the questions on work engagement were “yes,” “almost yes,” “somewhat indifferent,” and “no.” However, for the purposes of analysis, they were divided into two groups (high vs. low). Employees who answered “yes” and “almost yes” were classified into the high group, and those who answered “somewhat different” and “no” were classified into the low group.

### 2.3. Exposure Variables

Our exposure variables of interest were age, gender (male, female), marital status (unmarried, married, widowed/separated), number of members per household, physical disorders (shoulder pain, back pain), mental disorders (yes, no), healthy behavior (physical activity, sleeping hours, not eating breakfast, drinking alcohol, and smoking), interaction with supervisors/coworkers (worse, unchanged, better), working hours (less than 40 h/week, equal to or more than 40 h/week), type of job (office work, work in sales, and physical labor), workplace location, and socioeconomic status (SES). The SES variable included educational level (categorized into two groups: ≤12 years and >12 years of formal education) [16], income level (categorized by the household equivalent income [<JPY 2.99 million, JPY 3.0–7.99 million, >JPY 8.0 million, and unknown]) [16], and employee status (permanent, precarious, and other) [17].

### 2.4. Assessment of Confounders

The following factors were considered potential confounders between work engagement and working from home: age, gender, marital status, family number per household, shoulder/back pain, mental disorders, health behaviors (physical activity, sleeping hours, not eating breakfast, drinking alcohol, and smoking), interactions (with supervisors and coworkers), working hours, type of job, workplace location, and SES (education, household income, and employee status).

Previous studies have shown that higher scores in K6, a six-item self-report questionnaire designed to screen for mood and anxiety disorders, are correlated with low work engagement [18]. This may be because either low work engagement leads to more psychological distress, or alternatively, workers experiencing high distress are less likely to be engaged. Because the prevalence of depressive symptoms among adults during the COVID-19 pandemic is increasing [19], we considered mental disorders as a potential confounder.

In addition, previous research has demonstrated an interactive pathway between different health behaviors (physical activity, sleep, drinking and smoking behaviors, and dietary pattern), and between the change in health behaviors and work engagement [12,20,21]. Following the pandemic, new work styles have emerged that are affecting employees’ health behaviors; hence, we also considered health behaviors as potential confounders in this study.

According to the JR model, the demands and level of control associated with a job, and the support provided by one’s supervisor and colleagues are both associated with work engagement. Hence, we controlled for these job characteristics (interactions with supervisors and coworkers and working hours) as potential confounders [8].

### 2.5. Sample Characteristics

The study enrolled 28,000 participants. We excluded participants who were unemployed (*N* = 10,981) and who had missing data (*N* = 1349). The final analysis sample contained 15,670 workers (11,894 participants who worked from home and 3776 participants who worked but not from home).

## 3. Results

Table 1 summarizes the characteristics of the study subjects. In the analysis, continuous data were expressed as means (standard deviations) and categorical data as numbers (percentages). The question (“Do you feel energized when you are at work?”), which is a part of the New BJSQ, was used for the main analysis. The associations between employees’ work engagement and other covariates were estimated using the Poisson regression model because the outcome was more than 10% [22]. Further, the prevalence ratio (PR) and 95% confidence intervals (CI) for engagement were estimated after adjusting for age, gender, marital status, family number per household, shoulder/back pain, mental disorders, health behaviors (physical activity, sleeping hours, not eating breakfast, drinking alcohol, and smoking), interactions (with supervisors and coworkers), working hours, type of job, workplace location, and SES (education, household income, and employee status) (Table 2). Furthermore, the association of work engagement with the change in sleeping hours, interactions with supervisors, and working hours is shown in Figure 1, Figure 2 and Figure 3. Another question (“Do you take pride in your work?”), which is also part of the New BJSQ, was used for sensitivity analysis and for clarifying employees’ work engagement. Sensitivity analysis was also performed in a similar manner (Table 3). The associations between employees’ work engagement and other covariates, by whether employees were working from home or not, were estimated using the Poisson regression model, as shown in Appendix A.

A two-tailed *p*-value < 0.05 was considered as statistically significant. All analyses were performed using R, version 3.6.0 (R Foundation for Statistical Computing, Vienna, Austria).

Table 1 shows the characteristics of the study subjects. Among the 15,670 workers, 3776 were working from home at the time of the survey. Participants who were working from home were younger, had higher educational levels, and had higher incomes than those who were not working remotely. About 44% of all the participants had high work engagement. Further, sensitivity analysis revealed that 59.5% of all the participants had high work engagement.

In the univariate model, close communication with supervisors and colleagues, and working hours of <40 h/week, were associated with high work engagement in both groups (employees who worked from home and those who did not) (Table 2). In addition, Table 2 depicts the results of the multivariate analysis with an adjustment for potential confounders. Among those who worked from home, the absence (compared to the presence) of a mental disorder (PR, 1.33; 95% CI, 1.18–1.50, *p* < 0.001) and better (compared to worse) interactions with a supervisor (PR, 1.61; 95% CI, 1.17–2.22, *p* < 0.001) were independently associated with higher work engagement. However, unchanged opportunities (compared to more opportunities) for physical activity (PR, 0.86; 95% CI, 0.95–1.00, *p* = 0.04), unchanged and decreased (compared to increased) sleeping hours (PR, 0.81; 95% CI, 0.72–0.92, *p* < 0.001; PR, 0.72; 95% CI, 0.59–0.88, *p* < 0.001), unchanged opportunities (compared to increased opportunities) for eating breakfast (PR, 0.67; 95% CI, 0.5–0.9, *p* = 0.008; PR, 0.72; 95% CI, 0.59–0.88, *p* < 0.001), and working hours of ≥40 h/week (compared to working hours of <40 h/week) (PR, 0.84; 95% CI, 0.74–0.95, *p* = 0.01) were independently associated with lower engagement.

Among those who had increased their sleep duration, the proportion of participants with more work engagement was higher than those with low work engagement (*p* < 0.001) (Figure 1). Among those who had better interaction with supervisors, a larger proportion had higher work engagement (*p* < 0.001) (Figure 2). Among those who worked <40 h/week, the proportion of those who had high work engagement was larger than those who did not (*p* < 0.001) (Figure 3).

Table 3 depicts the results of the sensitivity analysis. Among those who worked from home, the absence (compared to the presence) of mental disorders (PR, 1.21; 95% CI, 1.10–1.34, *p* < 0.001), unchanged (compared to worse) interactions with a supervisor (PR, 1.25; 95% CI, 1.03–1.51, *p* = 0.02), and better (compared to worse) interactions with coworkers (PR, 1.4; 95% CI, 1.05–1.87, *p* = 0.02) were associated with higher engagement. The results of the sensitivity analysis support the main results.

Appendix A shows the PR and 95% CI for high work engagement by whether or not employees were working from home using the Poisson regression model. Among those who did not work from home, the absence (compared to the presence) of a mental disorder (PR, 1.34; 95% CI, 1.25–1.44, *p* < 0.001) and unchanged and better (compared to worse) interactions with a supervisor (PR, 1.37; 95% CI, 1.18–1.60, *p* < 0.001; PR, 1.76; 95% CI, 1.42–−2.68, *p* < 0.001) were independently associated with higher work engagement. In addition, working hours of ≥40 h/week (compared to working hours of <40 h/week) (PR, 0.83; 95% CI, 0.77–0.88, *p* < 0.001) were independently associated with lower engagement.

## 4. Discussion

This cross-sectional study found that close communication with superiors, refraining from working long hours, and obtaining adequate sleep are associated with high work engagement in Japanese employees working from home. To our knowledge, this is the first study to examine the factors associated with the work engagement of employees working from home in a large sample of adults. Behaviors or risks relating to lower work engagement were more prevalent in employees who worked from home than those who did not. This suggests that solving these problems is a viable target for interventions aimed at helping adults achieve high levels of work engagement.

Our study indicated that close communication with superiors and refraining from working long hours are the main factors related to the work engagement of employees working from home. These two factors are generally considered predictive of work engagement [8,9]. In addition, the reasons why these factors are associated with high work engagement among those working from home may be as follows: First, employees tend to focus on various aspects outside of work (e.g., looking after one’s children, cleaning up one’s room, and perhaps watching television) and have difficulty concentrating on their work, which causes them to multitask. Studies reveal that multitasking does not enable one to focus completely on a single aspect, resulting in diminished attention to detail and decreased opportunity to perform tasks at work [23]. Second, working from home decreases employees’ communication with their supervisors and colleagues. When work-related advice or assistance is needed, they find it more difficult to ask supervisors and colleagues for help at the beginning of the day. Hence, it is recommended that managers are frequently updated on the employee’s schedule, that managers examine what their employees are working on, and that weekly meetings are held [24]. All of these factors increase work duration and decrease work engagement. A review study of the mental and physical health effects of working at home showed that health outcomes such as well-being, stress, depression, and happiness were influenced by organizational support, colleague support, and social connectedness [25]. Our results corroborate such findings, because work engagement may be considered a positive psychological force in people’s lives.

Our study also implies that sleep behaviors may affect employees’ work engagement. Employees who recorded an increase in sleeping hours after starting to work from home had higher work engagement compared to others. A previous study indicated that individuals with poor sleep hygiene have lower self-regulatory capacity and lesser psychological strain and work engagement compared to those with high sleep hygiene [20]. Several studies support this evidence and have suggested a link between sleep and self-regulation/psychological strain relating to work engagement [26,27]. Although our study indicated that good sleep behavior is related to work engagement, we did not clarify the pathway through which good sleep behavior boosts work engagement among those working from home; this point should be examined further in future studies.

Additionally, this study has practical implications with respect to increasing the work engagement of employees working from home. The most significant struggle associated with working from home is maintaining productivity along with following healthy behaviors [28]. Further, the biggest challenges encountered by people working from home include difficulties in collaborating and/or communicating with colleagues and working longer hours, most of which are caused by the imbalance between employees’ professional and personal lives. In our study, more employees working from home shortened their sleep time than those not working from home. A short sleep duration is related to low productivity and, in turn, long working hours, which creates a vicious cycle. Based on the findings of earlier studies that high work engagement has beneficial effects on work performance and healthy behaviors [12,21,29], employers should strive to retain the productivity of their employees in accordance with the changing work style. For instance, to enable employees to maintain good interactions with their supervisors and colleagues, employees should be offered proactive online encouragement by superiors [19]. Some organizations conduct online lunch meetings with team members. Employees will be able to maintain a good relationship with their supervisor, be more productive at work, and, ultimately, not have to lose much sleep. In this study, these behaviors or risks, relating to lower work engagement, were found more in employees who were working from home compared to those who were not. Currently, evidence to clearly understand the effect of these challenges on work engagement is lacking and this is an important direction for future research. It is, however, certain that employees’ work engagement should be improved to overcome the aforementioned obstacles and increase productivity.

In this study, although similar behaviors or risks related to lower work engagement (worse communication with superiors, working long hours, and inadequate sleep) were indicated among those not working from home, these factors were found more frequently among employees who were working from home compared to those who were not. Specifically, having better communication with superiors, working reasonable hours, and getting an adequate amount of sleep may be a bigger challenge among those working from home. In view of this, workplaces that have implemented the work from home system should be more proactive in grasping these factors. It is desirable to create workplace environments that facilitate adaptation to a higher work engagement and healthier lifestyle among employees working from home.

Our study has several limitations. First, our findings cannot be generalized to suit other ethnic or age groups. Workers in Japan tend to report lower work engagement scores compared to those in other countries [30]. Second, due to the cross-sectional design of our study, we cannot exclude the possibility that causality runs in the opposite direction, suggesting that low work engagement causes an increase in working hours. Third, among the group not working from home, there was a mix of people who desired to work from home but could not and people who could not work from home due to their line of work. Although the results of the analysis must be carefully understood, our main goal was to indicate which factors were related to employees’ work engagement working from home. Finally, we could not adjust for a full set of potential confounders, such as company size, detailed work duties, corporate social responsibility (CSR), or organizational justice, all of which may affect work engagement. For instance, CSR, especially internal CSR, refers to the voluntary behaviors of corporations, such as treating their employees fairly, ensuring a good working environment, and providing employees with career development opportunities and facilities to improve their health condition, in addition to their organizational outcomes and behaviors [31]. External CSR refers to acts of social responsibility targeted toward a local community or the natural environment, specifically, donations for the protection of natural and cultural properties, development of educational programs, and local beautification activities [32]. Previous studies have implied a positive association between CSR and work engagement [33,34]. Further research should investigate how these factors affect the work engagement of employees working from home.

## 5. Conclusions

This study aimed to identify the factors that influence the work engagement of employees working from home during the COVID-19 pandemic in Japan. We found that close communication with superiors, refraining from working long hours, and obtaining adequate sleep are associated with high work engagement in Japanese workers working from home. This suggests that interventions aimed at addressing these factors may help workers achieve higher levels of work engagement.

## Figures and Tables

**Figure 1 ijerph-18-10495-f001:**
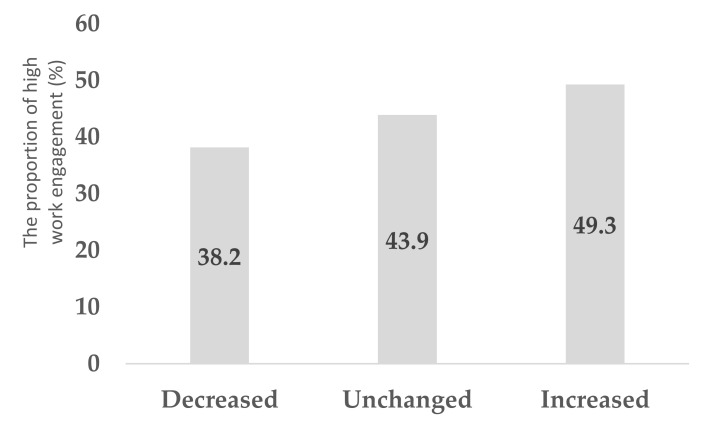
The association between the change in sleeping hours and higher work engagement among employees working from home.

**Figure 2 ijerph-18-10495-f002:**
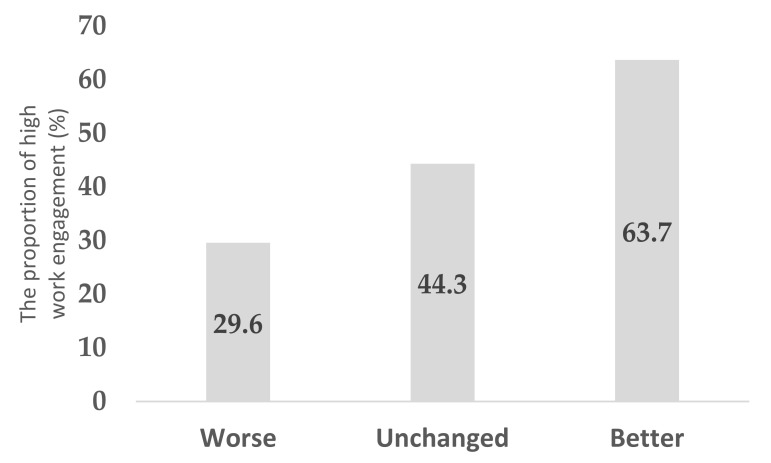
The association between the change in interaction with supervisors and higher work engagement among employees working from home.

**Figure 3 ijerph-18-10495-f003:**
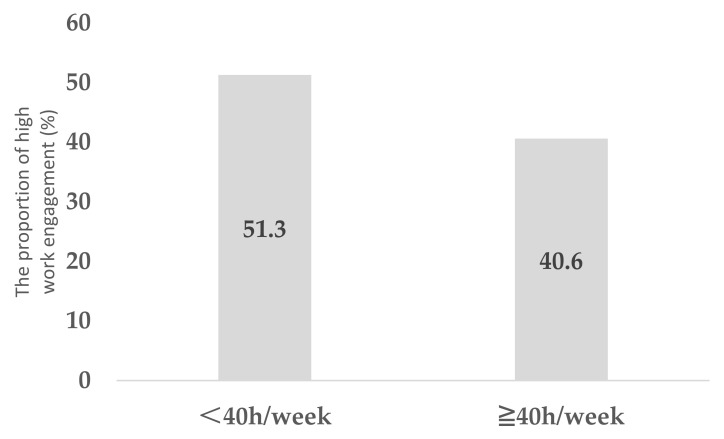
The association between working hours and higher work engagement among employees working from home.

**Table 1 ijerph-18-10495-t001:** Characteristics of study subjects according to their working from home status.

		All	Working from Home (−)	Working from Home (+)	
		*N* (%)	*N* (%)	*N* (%)	*p* Value
Total		15,670 (100.0)	N = 11,894	N = 3776	
Age		45.6 (13.8)	46.1 (13.9)	43.9 (13.1)	<0.001
Gender, male		9137 (58.3)	6598 (55.5)	2539 (67.2)	<0.001
Marital status	Married	9351 (59.7)	7010 (58.9)	2341 (62.0)	<0.001
	Never married	4926 (31.4)	3725 (31.3)	1201 (31.8)	
	Widowed	258 (1.6)	215 (1.8)	43 (1.1)	
	Separated	1135 (7.2)	944 (7.9)	191 (5.1)	
Number of members per household		2.8 (2.4)	2.8 (2.5)	2.7 (2.1)	0.13
Shoulder pain	Yes	7492 (47.8)	5451 (45.8)	2041 (54.1)	<0.001
Back pain	Yes	5817 (37.1)	4305 (36.2)	1512 (40.0)	<0.001
Mental disorders	Yes	5736 (36.6)	4209 (35.4)	1527 (40.4)	<0.001
Healthy behavior					
Physical activity	Decreased	4085 (26.1)	2754 (23.2)	1331 (35.2)	<0.001
	Unchanged	9972 (63.6)	8106 (68.2)	1866 (49.4)	
	Increased	1613 (10.3)	1034 (8.7)	579 (15.3)	
Sleeping hours	Decreased	1666 (10.6)	933 (7.8)	733 (19.4)	<0.001
	Unchanged	12,436 (79.4)	9809 (82.5)	2627 (69.6)	
	Increased	1568 (10.0)	1152 (9.7)	416 (11.0)	
Not eating breakfast	Increased	721 (4.6)	448 (3.8)	273 (7.2)	<0.001
	Unchanged	13,744 (87.7)	10,648 (89.5)	3096 (82.0)	
	Decreased	1205 (7.7)	798 (6.7)	407 (10.8)	
Drinking alcohol	Increased	1642 (10.5)	1065 (9.0)	577 (15.3)	<0.001
	Unchanged	10,871 (69.4)	8736 (73.4)	2135 (56.5)	
	Decreased	3157 (20.1)	2093 (17.6)	1064 (28.2)	
Smoking	Increased	667 (4.3)	423 (3.6)	244 (6.5)	<0.001
	Unchanged	13,720 (87.6)	10,561 (88.8)	3159 (83.7)	
	Decreased	1283 (8.2)	910 (7.7)	373 (9.9)	
Interacting					
With supervisors	Worse	1292 (8.2)	930 (7.8)	362 (9.6)	<0.001
	Unchanged	13,780 (87.9)	10,584 (89.0)	3196 (84.6)	
	Better	598 (3.8)	380 (3.2)	218 (5.8)	
With co-workers	Worse	1105 (7.1)	797 (6.7)	308 (8.2)	<0.001
	Unchanged	13,963 (89.1)	10,710 (90.0)	3253 (86.1)	
	Better	602 (3.8)	387 (3.3)	215 (5.7)	
Working hours	≤40 h/week	10,842 (69.2)	7810 (65.7)	3032 (80.3)	<0.001
Kind of job	Office work	7670 (48.9)	4988 (41.9)	2682 (71.0)	<0.001
	Work in sales	3842 (24.5)	2981 (25.1)	861 (22.8)	
	Physical labor	4158 (26.5)	3925 (33.0)	233 (6.2)	
Socioeconomic status					
Education	>12 y	11,771 (75.1)	8462 (71.1)	3309 (87.6)	<0.001
Household income	−2.99	2129 (13.6)	1858 (15.6)	271 (7.2)	<0.001
(million JPY/year)	3.00–7.99	7486 (47.8)	5763 (48.5)	1723 (45.6)	
	8.00-	3641 (23.2)	2274 (19.1)	1367 (36.2)	
	Unknown	2414 (15.4)	1999 (16.8)	415 (11.0)	
Employee status	Permanent	9837 (62.8)	6880 (57.8)	2957 (78.3)	<0.001
	Precarious	4277 (27.3)	3757 (31.6)	520 (13.8)	
	Other	1556 (9.9)	1257 (10.6)	299 (7.9)	
Work engagement *	high	6874 (43.9)	5204 (43.8)	1670 (44.2)	<0.001
Work engagement (sensitivity analysis) **	high	9327 (59.5)	6949 (58.4)	2378 (63.0)	<0.001

h, hour; y, year. In the analysis, continuous data were expressed as means (standard deviations) and categorical data as numbers (percentages). * The question (“Do you feel energized when you are at work?”) was used for the main analysis. ** Another question (“Do you take pride in your work?”) was used for sensitivity analysis and for clarifying employees’ work engagement.

**Table 2 ijerph-18-10495-t002:** Prevalence ratio and 95% confidence interval values for high work engagement among employees working from home using the Poisson regression model (univariable and multivariable analysis).

		Univariable			Multivariable		
		PR	95% CI	*p* Value	PR	95% CI	*p* Value
Age (per 10 years)		1.04	1.00–1.09	<0.001	1.04	1.00–1.09	0.01
Gender (male/female)		1.04	0.94–1.15	0.5	1.07	0.95–1.2	0.21
Marital status	Married	Reference			Reference		
	Never married	0.80	0.72–0.9	<0.001	0.85	0.75–0.97	0.02
	Widowed	1.39	0.95–2.02	<0.001	1.14	0.77–1.68	0.51
	Separated	1.00	0.81–1.24	0.97	0.97	0.84–1.17	0.73
Shoulder pain (Yes/No)		0.98	0.89–1.08	0.72	1.06	0.95–1.13	0.32
Back pain (Yes/No)		1.16	1.05–1.28	<0.001	1.06	0.94–1.17	0.34
Mental disorders (Yes/No)		1.43	1.29–1.58	<0.001	1.33	1.18–1.50	<0.001
Health behavior							
Physical activity	Increased	Reference			Reference		
	Unchanged	0.81	0.71–0.83	<0.001	0.86	0.75–1.00	0.04
	Decreased	0.81	0.71–0.84	<0.001	0.87	0.75–1.00	0.06
Sleeping hours	Increased	Reference			Reference		
	Unchanged	0.81	0.72–0.91	<0.001	0.81	0.72–0.92	<0.001
	Decreased	0.67	0.55–0.86	<0.001	0.72	0.59–0.88	<0.001
Not eating breakfast	Increased	Reference			Reference		
	Unchanged	0.87	0.73–0.91	0.12	0.67	0.5–0.9	0.008
	Decreased	1.03	0.83–1.28	0.8	0.9	0.63–1.26	0.53
Drinking alcohol	Increased	Reference			Reference		
	Unchanged	0.98	0.80–1.19	0.82	1.06	0.92–1.24	0.41
	Decreased	1.09	0.86–1.39	0.99	1.07	0.91–1.26	0.44
Smoking	Increased	Reference			Reference		
	Unchanged	0.98	0.82–1.18	0.86	1.02	0.82–1.26	0.89
	Decreased	0.89	0.72–1.11	0.3	1.02	0.79–1.32	0.65
Interacting							
With supervisors	Worse	Reference			Reference		
	Unchanged	1.49	1.23–1.81	<0.001	1.24	0.98–1.57	0.07
	Better	2.12	1.65–2.7	<0.001	1.61	1.17–2.22	<0.001
With co-workers	Worse	Reference			Reference		1.2
	Unchanged	1.49	1.21–1.89	<0.001	1.2	0.93–1.59	0.16
	Better	1.4	1.58–2.68	0.02	1.31	0.93–1.83	0.12
Working hours	<40 h/week	Reference			Reference		
	≥40 h/week	0.75	0.67–0.87	<0.001	0.84	0.74–0.95	0.01
Socioeconomic status							
Education (≤12 y/>12 y)		1.19	1.08–1.31	<0.001	0.97	0.84–1.13	0.73
Household income (million JPY/year)	−2.99	Reference			Reference		
	3.00–7.99	0.85	0.71–1.02	0.09	0.94	0.77–1.15	0.54
	8.00-	0.99	0.82–1.19	0.9	1.05	0.85–1.31	0.63
	Unknown	0.89	0.71–1.11	0.3	0.94	0.74–1.19	0.60
Employee status	Permanent	Reference			Reference		
	Precarious	1.06	0.93–1.22	0.19	0.98	0.84–1.15	0.85
	Other	1.47	1.26–1.71	<0.001	1.27	1.07–1.52	0.01

CI, confidence interval; h, hour; JPY, Japanese Yen; PR, prevalence ratio; y, year.

**Table 3 ijerph-18-10495-t003:** Results of sensitivity analysis: the prevalence ratio and 95% confidence interval values for high work engagement using the Poisson regression model (univariable and multivariable analysis).

		Univariable			Multivariable		
		PR	95% CI	*p* Value	PR	95% CI	*p* Value
Age (per 10 years)		1.04	1.00–1.09	<0.001	1.04	1.00–1.09	<0.001
Gender (male/female)		1.04	0.94–1.15	0.5	1.04	0.94–1.15	0.42
Marital status	Married	Reference			Reference		
	Never married	0.8	0.72–0.87	<0.001	0.91	0.84–1.02	0.09
	Widowed	1.39	0.95–2.02	0.09	1.05	0.74–1.48	0.80
	Separated	1.00	0.81–1.24	0.97	0.89	0.82–1.2	0.94
Shoulder pain (Yes/No)		1.19	1.08–1.31	<0.001	0.98	0.89–1.08	0.72
Back pain (Yes/No)		1.16	1.05–1.28	<0.001	1.04	0.94–1.14	0.41
Mental disorders (Yes/No)		1.43	1.29–1.58	<0.001	1.21	1.10–1.34	<0.001
Health behavior							
Physical activity	Increased	Reference			Reference		
	Unchanged	0.81	0.71–0.93	<0.001	0.93	0.82–1.05	0.22
	Decreased	0.81	0.71–0.99	<0.001	0.96	0.85–1.09	0.56
Sleeping hours	Increased	Reference			Reference		
	Unchanged	0.81	0.72–0.91	<0.001	0.91	0.82–1.02	0.1
	Decreased	0.67	0.55–0.81	<0.001	0.96	0.76–1.04	0.14
Not eating breakfast	Increased	Reference			Reference		
	Unchanged	0.87	0.73–1.04	0.12	1.01	0.85–1.20	0.88
	Decreased	1.03	0.83–1.28	0.8	1.03	0.84–1.26	0.71
Drinking alcohol	Increased	Reference			Reference		
	Unchanged	1	0.87–1.14	0.96	0.95	0.84–1.07	0.39
	Decreased	1.01	0.89–1.18	0.9	1.02	0.89–1.16	0.8
Smoking	Increased	Reference			Reference		
	Unchanged	0.98	0.80–1.19	0.82	0.98	0.82–1.18	0.86
	Decreased	1.09	0.86–1.1	0.49	0.89	0.72–1.11	0.3
Interacting							
With supervisors	Worse	Reference			Reference		
	Unchanged	1.49	1.23–1.82	<0.001	1.25	1.03–1.51	0.02
	Better	2.12	1.65–2.72	<0.001	1.21	0.92–1.61	0.7
With co-workers	Worse	Reference			Reference		
	Unchanged	1.49	1.21–1.89	<0.001	1.1	0.9–1.35	0.36
	Better	2.06	1.58–2.69	<0.001	1.4	1.05–1.87	0.02
Working hours	<40 h/week	Reference			Reference		
	≥40 h/week	0.78	0.67–0.87	<0.001	0.94	0.84–1.05	0.3
Socioeconomic status							
Education (≤12 y/>12 y)		0.91	0.79–1.05	0.09	1.02	0.9–1.16	0.72
Household income (million JPY/year)	−2.99	Reference			Reference		
	3.00–7.99	0.97	0.83–1.15	0.76	1.01	0.85–1.2	0.91
	8.00-	1.10	0.93–1.29	0.81	1.09	0.9–1.31	0.62
	Unknown	0.93	0.76–1.13	0.96	0.91	0.74–1.12	0.37
Employee status	Permanent	Reference			Reference		
	Precarious	1.06	0.93–1.22	0.39	0.97	0.85–1.11	0.69
	Other	1.47	1.26–1.71	<0.001	1.22	1.05–1.42	0.01

CI, confidence interval; h, hour; JPY, Japanese Yen; PR, prevalence ratio; y, year. Note: The results are adjusted for age, gender, marital status, family number per household, shoulder/back pain, mental disorders, health behaviors (physical activity, sleeping hours, not eating breakfast, drinking alcohol, and smoking), interactions (with supervisors and coworkers), working hours, type of job, location, and socioeconomic status (education, household income, and employee status). Another question (“Do you take pride in your work?”) that is also a part of the New BJSQ was used for sensitivity analysis and for clarifying employees’ work engagement.

## Data Availability

The data used in this study are not available in a public repository because they contain personally identifiable or potentially sensitive patient information. Based on the regulations for ethical guidelines in Japan, the Research Ethics Committee of the Osaka International Cancer Institute has imposed restrictions on the dissemination of the data collected in this study. All data enquiries should be addressed to the person responsible for data management, Takahiro Tabuchi at the following e-mail address: tabuchitak@gmail.com.

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
