# Peer review of "Factors Associated with the Work Engagement of Employees Working from Home during the COVID-19 Pandemic in Japan"

_ijerph, 2021, doi:10.3390/ijerph181910495_

Round 1

Reviewer 1 Report

It is an excellent work, well presented, easy to read and systematic to lead the reader in the development of the research. However, if it is considered as a further limitation to the present study, the indicators of work engagement could be broader, due to its multidimensional nature (cognitive, physical and emotional). The work shows a strong consistency with the findings obtained in other studies, but these are reduced part of the references used, it is suggested to review and add more works in this regard (the results in other populations are left to the judgment of the authors to consider whether such findings are relevant).

Reviewer 2 Report

It is a very good articel, which deals with an extremely topical topic. The forced transition to remote work emphasized all its advantages and disadvantages, but also in many areas dispelled the concerns of employers. These concerns concerned, inter alia, commitment to work. The presented research is important because their implications also apply to post-pandemic time and may positively affect the wider use of remote work in the organization's activities.

It is useful to illustrate the results graphically as well. 

Reviewer 3 Report

First, I would like to thank you for the opportunity to read this work. This is an interesting study about a current topic – i.e., identify factors that may influence the work engagement of employees working from home during the COVID-19 pandemic. I am also impressed with the sample collected of 15,670 individuals. Moreover, since most of the studies are performed in western countries it is very interesting to have data from Japan.

Below I leave some comments and suggestions. Hope they contribute to the improvement of the paper.

(1)       In the introduction section the authors could highlight more the main goal of the current study and the expected theoretical and practical contributions.

(2)       In the “Exposure measures” section, was not clear to me how each variable was assessed. More precisely, how gender, mental disorders, interaction with supervisors/coworkers, and working hours were measured? Include this information, please.

(3)       While reading the paper I felt sometimes the information provided was repetitive. In addition, since data were also collected with individuals not working from home, I believe the data collected could be more deeply explored by the authors. 

(4)       The Materials and Methods section and the Results section needs to be more detached from each other. In other words, the authors should detach the information that belongs to the sample characteristics (which in my opinion would be better provided in the material and methods section, as the authors did) and the data analysis used to achieve the research goals (which in my opinion should be presented in the results section, including the tables).

Round 2

Reviewer 3 Report

I would like to congratulate the authors for the improvements in this new version of the paper. I have nothing more substantial to add. I just have a little note: please, write "job demands-resources model (JD-R model)" (in the introduction section, line 42).